# Immunohistochemical Characterization of Feline Giant Cell Tumor of Bone (GCTb): What We Know and What We Can Learn from the Human Counterpart

**DOI:** 10.3390/ani15050699

**Published:** 2025-02-27

**Authors:** Ilaria Porcellato, Giuseppe Giglia, Leonardo Leonardi

**Affiliations:** Department of Veterinary Medicine, University of Perugia, 06121 Perugia, Italy; ilariaporcellatodvm@gmail.com (I.P.); leonardo.leonardi@unipg.it (L.L.)

**Keywords:** feline, giant cell tumor of bone, immunohistochemistry, osteoclast-like cells, RUNX2, Karyopherin α2, IBA1, TRAP, RANK

## Abstract

Giant cell tumor of bone (GCTb) is a benign tumor in human medicine, still, in veterinary medicine, where it is more commonly described in cats, its recognition and diagnosis are still a challenge. With this study, we provide new insights into the histological and immunohistochemical phenotype of the tumor, confirming similarity with the human tumor, and encouraging further studies on this neoplastic entity also in our pets.

## 1. Introduction

In humans, giant cell tumor of bone (GCTb, also known as osteoclastoma) is a primary benign tumor of bones, that displays locally aggressive behavior, with high local recurrence rates; also, it can occasionally progress to a malignant form [1]. This tumor typically arises in individuals with complete skeletal maturation and mainly affects long bone epiphyses, causing deep alteration to bone structure and impacting joint functions. GCTb is the most common benign tumor of bones in humans representing up to 20% of benign primary bone tumors and 4–10% of the total primary bone tumors [1,2,3].

GCTbs are reported to affect all ethnical groups, but a higher prevalence (up to 20% of primary bone tumors) has been reported in Chinese patients [4] and in India (20.3% of primary bone tumors [5]), where the incidence of GCTBs is higher than in the Western populations [6]. Recent data collection from the suburban New York healthcare system revealed 4.9% GCTb diagnoses out of total benign bone lesions, with osteochondroma representing the most common, with 47% of the total number of diagnoses [7].

Several studies report a slight predominance of this diagnosis in women over men, particularly in Western countries [8], with variable percentages [9] (54.6% women and 5.4% men reported in a Brazilian study [10]; 58% women and 42% men in a Dutch study from Pathology Registry [11]; 54% women and 46% men in a report from the Swedish Cancer Registry) and a female-to-male ratio ranging from 1:1.1 to 1:1.5 [3]. On the other hand, other studies indicate a higher prevalence in men, with a male/female sex ratio of 1.27–1.77:1 [8].

It is commonly reported in young adults, aged between 20 and 40 years, but there are reports of this tumor also for patients > 50 years. Some studies report a second peak in diagnoses among 50–59-year-olds.

Histologically, GCTbs are typically composed of a striking number of multinucleated osteoclast-like giant cells intermingled with a stromal mononuclear population [12], partially composed of macrophages [13,14]. In human medicine, giant multinucleated cells have been demonstrated to be reactive osteoclasts [15]. The mononuclear stromal cell population is supposed to be the neoplastic and proliferative GCTb component, inducing the massive osteoclast-like cell differentiation. This cellular population is composed of stromal cells and mononuclear monocyte cells, considered to be either reactive macrophages (tumor-associated macrophages, TAMs) or osteoclast precursors [16]. Some studies demonstrated that the stromal cells expressed many osteoblastic markers and showed properties of pre-osteoblast-like cells, leading to the hypothesis that GCTb is of osteoblast lineage origin [17]; some other studies supported a mesenchymal stem cell origin of the lesion.

Immunohistochemically, human GCTb shows mononuclear stromal cells strongly positive for SATB2 and RUNX2, while the giant cell component is negative for both markers [18,19,20].

In humans, the presence of a driver mutation in the histone 3.3 (H3.3) gene *H3F3A* is the key to confirming the diagnosis, being described in more than 90% of cases [21,22]. These studies, to the best of the authors’ knowledge, have never been conducted on domestic animals or in cats.

In domestic animals, particularly in dogs and cats, reports on this type of tumor are rare, being mostly recognized in the feline population [23,24,25,26,27]. As a consequence, data on epidemiology, tumor biology, molecular characterization, and therapeutic approaches are very limited.

Indeed, in veterinary medicine, the prevalence of this tumor has never been reported, most of the cases described in scientific literature being single case reports [23,24,25,28,29] or small case series [26]. Nevertheless, it appears that GCTb is more commonly diagnosed in adult cats rather than dogs [27]. In the archives of our Pathology Service (Department of Veterinary Medicine, University of Perugia, Italy), in a period that goes from 2008 to 2024, a diagnosis of GCTb was reported in <1% of the total bone lesions of cats. Case reports of GCTb in dogs are even more sporadic in veterinary medicine scientific literature [30,31]. To support this, no diagnosis of GCTb has been reported in the Canine Cancer Registry of Umbria Region (Italy), on a total of >18,500 diagnoses recorded from October 2013 to September 2024 [32].

The histological diagnosis of GCTb is based on features that are similar to the human counterpart; a neoplasm composed of a large number of multinucleated giant cells often defined as “osteoclast-like”, intermingled with mononuclear round to oval cells, recognized as the real neoplastic component of the lesion, and a third cellular population of macrophages/monocytes, probably representing the osteoclast precursors [14,15]. Immunohistochemically, feline GCTb have been described for the mononuclear stromal cell component, positivity for osteoblast markers (i.e., osterix), and for the giant cell component, the expression of macrophage markers (i.e., Iba1) [25].

Different authors report that, in domestic animals, the main challenge is the differentiation with giant cell-rich osteosarcoma, particularly when based only on histological and cytological features [27,33]. As differential features, in giant cell-rich osteosarcoma, cellular anisocytosis, and anisokaryosis are expected to be more evident and associated with a variable deposition of osteoid matrix [33]. Moreover, imaging characteristics can be valuable in the diagnostic process; however, the limited number of studies that examine both clinical and pathological features in tandem makes it challenging to identify imaging features that strongly suggest a diagnosis of GCTb in pets.

GCTb is a very rare tumor in cats, and collecting cases is a significant challenge. Additionally, the immunohistochemical characterization of GCTB is hardly addressed in the literature. Therefore, descriptions of additional feline cases would benefit both cats and humans in assessing the immunohistochemical features of the cell populations of feline GCTb and evaluating similarities that could improve patient care. The present short case series aims at providing a histological and phenotypical description of three feline GCTBs, comparing our results with the data available for the human counterpart.

## 2. Materials and Methods

### 2.1. Case Selection

Cases of tumors compatible with GCTB were retrieved from the archive of the Department of Veterinary Medicine of the University of Perugia (Italy), starting in 2010. All samples represented routinely processed Formalin-Fixed Paraffin-Embedded (FFPE) tissues of selected cases representing biopsy specimens submitted to the laboratory for routine diagnostic.

Criteria for the inclusion in the case series were:-Clinical presentation of a monostotic, circumscribed expansile primary bone neoplasia with osteolysis [23,25,34] in a skeletally mature domestic cat (felis catus);-Neoplastic mononuclear stromal cell with mild cellular atypia (but mitoses can be numerous), together with numerous multinucleated giant cells;-Scant/absent osteoid deposition [35].

### 2.2. Histology, Histochemistry, and Immunohistochemistry

Hematoxylin and eosin (H&E) histological slides were re-evaluated on light microscopy by three pathologists (LL, GG, IP) to confirm the initial diagnosis using an Olympus BX50 microscope. Von Kossa stain was performed to evaluate the presence of mineralized matrix and bone spicules in the examined samples.

To perform the immunohistochemical characterization, 5 μm sections were cut using a Thermo Scientific Shandon Finesse ME microtome and mounted on poly-L-lysine-coated slides from formalin-fixed and paraffin-embedded samples, which were then dewaxed and dehydrated. Immunohistochemistry was performed on serial sections with antibodies raised against Iba1 [36], TRAP, SATB2, RUNX2 [37], RANK, karyopherin α2 (KPNA-2) [37], and osteocalcin to evaluate the expression of osteoblast and macrophagic markers in the various population of GCTb. Additionally, Ki-67 was used as a proliferation marker, as commonly used for the human GCTb [20,38,39]. Immunohistochemistry was performed following the protocols reported in Table 1. The incubation with the secondary antibody (Abcam, Cambridge, UK, ab93697 Mouse and Rabbit Specific HRP Plus ABC detection IHC kit was performed following manufacturer guidelines. Positivity was detected through a solution of aminoethyl carbazole (AEC). Carazzi’s hematoxylin was used as a counterstain. Coverslips were mounted with Aquatex® (Merck, Darmstadt, Germany). Positive controls were obtained from canine reactive lymph nodes for Iba1 and Ki-67 antibodies, whereas for TRAP, SATB2, RUNX2, KPNA-2, and osteocalcin, normal bone and osteosarcoma were used. Negative controls were run omitting the primary antibody and incubating control sections with TBS. Positivity for the characterization markers (Iba1, TRAP, SATB2, RUNX2, RANK, KPNA-2, Osteocalcin) was assessed by three pathologists (LL, GG, IP) and reported as “−” when the examined population completely lacked immunoreactivity; as “+/−” when < than 50% of the examined population showed immunoreactivity; as “+” when > than 50% of the examined population showed immunoreactivity.

Ki-67 index was calculated with QuPath (v0.5.0) on a single field (FN22, ×400) image captured with a digital camera (Nikon DS-Fi1, Tokyo, Japan). To obtain the percentage of positive nuclei, a full image annotation was created followed by the positive cell detection analysis. Cell detection analysis parameters were as follows: “detectionImageBrightfield”: “Optical density sum”, “background radius”: 35 px, “median filter radius”: 2.0 px, “sigma”: 2.0 px, “minimum area”: 10 px^2^, “maximum area”: 400.0 px, “Threshold”: 0.18, “maxBackground”: 1, “cell expansion”: 5.0 px, “include cell nucleus”: true, “smooth boundaries”: true, “make measurements”: true, “threshold compartment”: “Nucleus: DAB OD mean”, “thresholdPositive1”: 0.6, “thresholdPositive2”: 0.6, “thresholdPositive3”: 0.7, “singleThreshold”: true.

## 3. Results

### 3.1. Case Selection and Histological Features

From our archive, we selected three feline cases that met the inclusion criteria. The signalment and anamnesis of the three cats are reported in Table 2.

Histologically, the examined sections showed a well-demarcated, expansile, highly cellular proliferative process characterized by three distinctive cellular populations. The first one was represented by mononucleated, oval to spindle-shaped stromal cells with indistinct cell borders, a scant amount of eosinophilic cytoplasm, and an oval nucleus with irregularly dispersed chromatin and nucleoli. These cells exhibited mild atypia, while the mitotic count was high (20 to 43 mitotic figures on 2.37 mm^2^; FN 22, ×400). Stromal cells were supported by a scant fibrovascular stroma. Intermingled with the stromal cells were a high number of macrophages and numerous and large cells (up to 150 μm) with abundant homogeneously eosinophilic cytoplasm and multiple nuclei (up to >50), defined as osteoclast-like multinucleated giant cells, evenly distributed through the lesion (Figure 1a). At the periphery of the tumor growth, there were multifocal small areas of hemorrhage associated with macrophages containing intracytoplasmic ocher pigment (hemosiderin). Only rarely, bone spicules (grayish in color after Von Kossa staining) and scant amounts of extracellular homogeneous eosinophilic matrix were detected between cells.

### 3.2. Immunohistochiemistry

Neoplastic stromal cells showed diffuse nuclear positivity for RUNX2, SATB2, KPNA2 (Figure 1b–d). Multinucleated giant cells and occasional stromal cells showed a diffuse cytoplasmic positivity for TRAP (Figure 1e), while only multinucleated giant cells showed positivity for RANK, and Iba1 (Figure 1f,g), all characterized by a finely granular cytoplasmic reaction.

Multinucleated giant cells were invariably negative for SATB2, RUNX2, and KPNA2. Detailed results of the immunohistochemical characterization of the three cases are reported in Table 3. Additionally, neoplastic stromal cells but not multinucleated giant cells were rarely osteocalcin-positive. This was instead visibly positive in remodeled bone spicules and scant and rare deposits of extracellular matrix (Figure 1h).

Regarding the proliferative index (Ki-67), the nuclear positivity ranged from 0 to 8% of neoplastic mononuclear stromal cells (Figure 1i).

## 4. Discussion

Giant cell tumor of bone (GCTb) is a rare entity in veterinary medicine, that apparently shares numerous similarities with the human counterpart.

Multinucleated giant cells within the lesion showed a marked cytoplasmic positivity for IBA1, tartrate-resistant acid phosphatase (TRAP), and a moderate immunolabeling for RANK.

IBA1 is a marker commonly used in dogs and cats for the identification of cells of the monocytic/macrophagic origin [40] since CD68, which is considered a pan-macrophagic marker in humans, is not suitable for canine or feline tissues, particularly when formalin-fixed and paraffin-embedded [36]. Multinucleated giant cells in feline GCTB were invariably IBA1-positive, similarly to what reported in humans, where CD68 was investigated. Moreover, this result has been observed also by Carrete et al. in a feline vertebral GCTB recently described [25]. Within the tumor analyzed also occasional intratumoral mononucleated cells, interpreted as histiocytes, were present, as described also in the human counterpart [40].

Similarly to our results, TRAP expression was assessed in multinucleated giant cells and a subpopulation of mononucleated cells in different studies on human GCTb [41,42,43]. This peculiar TRAP pattern of expression has been hypothesized to be indicative of a monocytic osteoclast precursor phase of differentiation of giant cells [42]. Interestingly, after therapy with denosumab, that is used in humans for the treatment of unresectable GCTb, TRAP-positive cells decreased and became undetectable, suppressing tumor activity via inhibition of the RANK-RANKL pathway [41].

RANK pathway has been associated with the pathogenesis of GCTb in humans as it is involved in an imbalance between bone formation and its resorption, hence being involved in the osteolytic nature of the tumor [44]. Also, expression of RANK on osteoclast-like giant cells has been demonstrated in human GCTb and also in occasional mononuclear cells [45], similarly to what we observed in our cases.

Taken together, these results support an osteoclastic origin of the multinucleated cellular population within the tumor also in feline GCTBs, supporting their similarity with the human form.

Furthermore, all our cases were characterized by a diffuse nuclear SATB2 expression in most of the mononuclear cells, with multinucleated giant cells that were invariably negative for the marker. Also, this result mirrors what has previously been reported in human medicine, where Amzajerdi and Coll suggest that, whenever a positivity of multinucleated cells should be observed, a diagnosis of osteosarcoma should be favored [18].

RUNX2 is a transcription factor which is pivotal for osteoblast differentiation [46]. The nuclear expression of this protein has been described also in canine osteosarcoma [47] and extra-skeletal osteosarcomas, where it was positive only in the osteoblastic component, being invariably negative in osteoclast-like giant multinucleated cells [37]. This is similar to what we observed in GCTb, supporting the hypothesis that at least a part of the stromal cells is likely originating from the osteoblastic lineage. Moreover, in human GCTb, RUNX2 expression is associated with the upregulation of MMP13, which is the main proteinase expressed by the stromal cell component of the tumor [48]. Hence, the expression of this transcription factor could be implicated also in the prognosis and clinical behavior of the tumor, which could be assessed also in cats, including in future studies tumors with a complete follow-up.

Karyopherin α2 (KPNA-2) is a transport protein that mediates the nuclear translocation of numerous target proteins through the nuclear pore complex [49]. The expression of this protein has been associated with prognosis in different types of tumors, but, interestingly, has been also demonstrated differentially expressed in osteosarcoma and other bone tumors, such as chondrosarcoma and Ewing sarcoma [50]. The authors report negativity in different benign bone lesions, but GCTbs were not included in their study.

The positivity of most of the mononuclear cells, where also most of the Ki-67 positive nuclei were observed, support the hypothesis of an osteoblastic origin of the stromal component of the tumor. To support this hypothesis, there is also evidence of SATB2 expression, which has been described also in a significant number of human GCTb [18].

This study provides additional features on the immunohistochemical characteristics of the cell populations in feline GCTb providing data that could improve the differentiation between GCTb and lesions with similar features (i.e., giant cell osteosarcoma). Additionally, the expression of some of the molecules identified in this study could provide a potential starter point for future targeted therapies. For instance, the pathway RANK/RANKL has been identified as having a key role in the pathogenesis of GCTb and targeted by novel therapies [51]. 

Unfortunately, the main limit of this study is the lack of established diagnostic criteria for GCTB in cats and the impossibility of definitively confirming the diagnosis. Our small case series was obtained by borrowing information from the human counterpart. Nevertheless, we are completely aware that knowledge of this peculiar entity is still scant and would need the support of clinical and biomolecular data.

## 5. Conclusions

Currently, the diagnostic criteria for GCTBs in cats and domestic animals are still lacking, leading to a possible misdiagnosis that may suggest the worst prognosis in cats affected by osteoclast-rich bone tumors. Larger case series, including follow-up information and comprehensive diagnostic imaging of the lesions and the patient, associated with histopathology, phenotyping, and biomolecular analyses, are needed to enhance the clinicopathological diagnostic capability of identifying this unique oncological entity.

## Figures and Tables

**Figure 1 animals-15-00699-f001:**
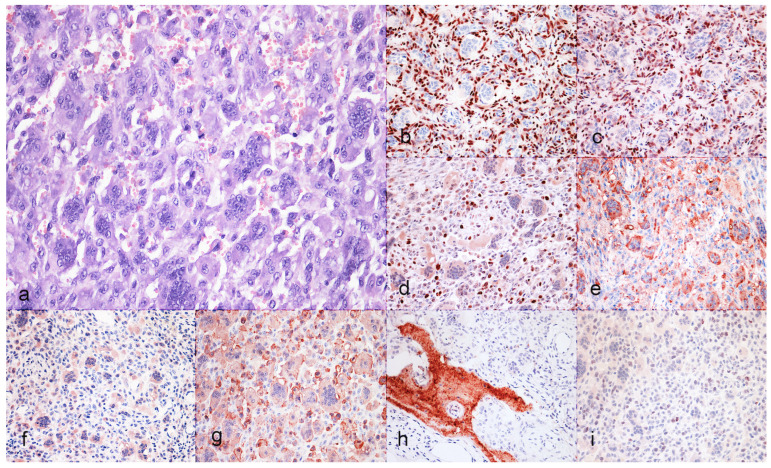
Histological (H&E; ×400) and immunohistochemical (×400) features of feline GCTb. (**a**) Cat, tibia. Histologically, the proliferative process is characterized by neoplastic stromal cells with frequent mitotic figures, macrophages and multinucleated giant cells (MGC); (**b**) neoplastic stromal cells show nuclear positivity for RUNX2 while MGC are diffusely negative; (**c**) neoplastic stromal cells, but not MGC show diffuse nuclear positivity for SATB2; (**d**) neoplastic stromal cells with intense nuclear positivity for KPNA2; (**e**) TRAP in MGC and neoplastic stromal cells; (**f**) MGC positivity for RANK; (**g**) Iba1 in macrophages and MGC; (**h**) osteocalcin expression in rare stromal cells and in bone spicules; (**i**) Ki-67 positive nuclei in neoplastic stromal cells. Counterstain was performed with Carazzi’s hematoxylin. Positive reaction is represented by red/brownish immunolabeling.

**Table 1 animals-15-00699-t001:** Antibodies and protocols for the immunohistochemical characterization of the tumors.

Antibody	Manufacturer	Dilution	Antigen Retrieval
Iba1	Merck Millipore	1:100	HIER, Tris-EDTA buffer; pH 9.0
TRAP	Santa Cruz Biotechnology	1:50	HIER, Tris-EDTA buffer; pH 9.0
SATB2	Cell Signaling Technology	1:200	HIER, Tris-EDTA buffer; pH 9.0
RUNX2	Santa Cruz Biotechnology	1:200	HIER, Tris-EDTA buffer; pH 9.0
RANK	Santa Cruz Biotechnology	1:50	HIER, Tris-EDTA buffer; pH 9.0
KPNA-2	Santa Cruz Biotechnology	1:150	HIER, Tris-EDTA buffer; pH 9.0
Osteocalcin	BioGenex LifeSciences	1:50	HIER, Tris-EDTA buffer; pH 9.0
Ki-67	Agilent Dako	1:200	HIER, Tris-EDTA buffer; pH 9.0

**Table 2 animals-15-00699-t002:** Signalment and tumor location of the three selected cases of GCTb.

Case	Breed	Age	Sex	Tumor Location
1	Domestic shorthair	15	M	Tibia
2	Domestic shorthair	15	F	Tibia
3	Siamese	5	M	Dewclaw

**Table 3 animals-15-00699-t003:** Results of the immunohistochemical analysis on mononucleated cells (MC: stromal neoplastic cells and macrophages) and multinucleated giant cells (MGC: osteoclast-like cells). Immunoreactivity is reported as “−” when the examined population completely lacked immunoreactivity; “+/−” when < than 50% of the examined population showed immunoreactivity; as “+” when > than 50% of the examined population showed immunoreactivity.

Case	Iba1		TRAP		SATB2		RUNX2		RANK		KPNA-2		Osteocalcin	
	MC	MGC	MC	MGC	MC	MGC	MC	MGC	MC	MGC	MC	MGC	MC	MGC
1	+/−	+	+/−	+	+	−	+	−	+/−	+	−	+	+/−	−
2	+/−	+	+/−	+	+	−	+	−	+/−	+	−	+	−	−
3	+/−	+	−	+	+/−	−	+	−	+/−	+	−	+	+/−	−

## Data Availability

The original contributions presented in this study are included in the article. Further inquiries can be directed to the corresponding author.

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
