# Peer review of "Immunohistochemical Characterization of Feline Giant Cell Tumor of Bone (GCTb): What We Know and What We Can Learn from the Human Counterpart"

_animals, 2025, doi:10.3390/ani15050699_

Round 1

Reviewer 1 Report

Comments and Suggestions for Authors

the manuscript appears well-developed, with a clear structure and precise description of methodology and results. However, here are some suggestions to ensure the final version is ready for submission:

1. Abstract: It is well-written. However, to enhance it further, consider adding a concise statement on the clinical or diagnostic significance of the study for veterinarians.

2. Introduction: Comprehensive, but you could better emphasize the comparative relevance between veterinary and human medicine in diagnosing these tumors, highlighting the current knowledge gap.

3. Table 3 (immunohistochemical results): Everything looks clear, but ensure all symbols (e.g., “+/-”) are defined in the table legend for clarity.

4. Figures: The histological and immunohistochemical images are illustrative, but double-check the resolution and labels to meet the journal’s guidelines.

5. Discussion: Well-structured, but adding a brief paragraph on potential therapeutic implications or directions for future research could strengthen the impact of your findings.

Author Response

Comments and Suggestions for Authors

the manuscript appears well-developed, with a clear structure and precise description of methodology and results. However, here are some suggestions to ensure the final version is ready for submission:

  1. Abstract: It is well-written. However, to enhance it further, consider adding a concise statement on the clinical or diagnostic significance of the study for veterinarians.

Author(s) response: This has been added in the manuscript.

  1. Introduction: Comprehensive, but you could better emphasize the comparative relevance between veterinary and human medicine in diagnosing these tumors, highlighting the current knowledge gap.

Author(s) response: This has been added in the manuscript.

  1. Table 3 (immunohistochemical results): Everything looks clear, but ensure all symbols (e.g., “+/-”) are defined in the table legend for clarity.

Author(s) response: This has been added in the table legend.

  1. Figures: The histological and immunohistochemical images are illustrative, but double-check the resolution and labels to meet the journal’s guidelines.

Author(s) response: This has been checked.

  1. Discussion: Well-structured, but adding a brief paragraph on potential therapeutic implications or directions for future research could strengthen the impact of your findings.

Author(s) response: This has been added in the manuscript.

Reviewer 2 Report

Comments and Suggestions for Authors

Dear authors

Although the manuscript is well written and structured, there are a few comments I would like you to take into account:

1) 3 cases is really too few for this work to be considered an original article. I strongly recommend that it be turned into a short communication.

2) Are all the antibodies validated for the species by western blot? If not, how were they validated?

3) Line 139: the authors mention that for the negative control, they chose to omit the primary antibody. I would like the authors to justify why they used this type of negative control, given that, according to the latest guidelines, this type of negative control is not recommended/valid. It would be important to discuss the limitations and advantages of this method and what it allows/does not allow us to conclude (please take a look in the article:

“Controls for Immunohistochemistry: The Histochemical Society's Standards of Practice for Validation of Immunohistochemical Assays” DOI: 10.1369/0022155414545224" 

Author Response

Comments and Suggestions for Authors

Dear authors

Although the manuscript is well written and structured, there are a few comments I would like you to take into account:

1) 3 cases is really too few for this work to be considered an original article. I strongly recommend that it be turned into a short communication.

Author(s) response: We thank the reviewer for his/her/their time in revising the ma manuscript. We are aware that three cases are few, but this pathological entity is quite rare. Moreover, while this manuscript presents only three cases, the comprehensive immunophenotyping provides significant insights that we believe justify its presentation as a full-length manuscript. We respectfully defer to the Editor's discretion on the final format.

2) Are all the antibodies validated for the species by western blot? If not, how were they validated?

Author(s) response: For each antibody not previously tested in veterinary medicine, we obtained the amino acid (aa) sequence of the antigen against which the antibody was raised. We then performed a sequence alignment between the canine protein and the species in which the antibody was originally validated by the manufacturer. This analysis was conducted on all the antibodies used in this study using the free online tools available on www.uniprot.it.

Additionally, for each antibody we tested different control tissues and tumors and compared the labelling pattern with the images reported for healthy and pathological tissues provided on www.proteinatlas.com

3) Line 139: the authors mention that for the negative control, they chose to omit the primary antibody. I would like the authors to justify why they used this type of negative control, given that, according to the latest guidelines, this type of negative control is not recommended/valid. It would be important to discuss the limitations and advantages of this method and what it allows/does not allow us to conclude (please take a look in the article:

Author(s) response: We thank the reviewer for the comment and the suggestion. We acknowledge that this type of control does not provide a good control for the primary antibody possible non-specific binding, but instead is, at best, a negative control for the secondary antibody. However, during our protocol assessment, we included different tissues that do not express the target antigens, such as unrelated tumors and other healthy control tissue from the same species (in this case, Felis catus).

In this case, in all tissue tested and expected to be negative, no positive immunolabeling was observed, supporting the specificity of the antigen. In all tissues tested that were expected to be negative, no positive immunolabeling was observed. This finding supports the specificity of the antibody for its target antigen.

Reviewer 3 Report

Comments and Suggestions for Authors

In this manuscript titled “Immunohistochemical characterization of feline giant cell tumor of bone (GCTb): what we know and what we can learn from human counterpart”, the authors analyze the expression of human GCTb marker proteins. This study of GCTb in cats provided interesting data.

The data presented in this study are quite preliminary. The experimental results presented in this manuscript are not clearly described and it is unclear whether conceptual advances can be achieved. Overall, the manuscript is too preliminary to be considered for publication. 

Comments:

1.     Line 25-26: The authors should select and list GCTb marker proteins. Ki-67 is used as a proliferation marker for GCTb and should be removed.

2.     Line 26-30: The study merely observed the expression of marker proteins in specific tissues. The reviewer does not know why that is sufficient to conclude that they are comparable to humans. The authors should logically describe what they found in this study.

3.     Intro: The reasons for studying GCTb in cats with a small number of cases are not compelling. The authors should focus their statement on this point.

4.     Mate & Meth, line 126-144: How did you obtain the cat samples? How was the tissue fixed and under what conditions? What type of equipment was used to prepare the sections? What kind of microscope did you use? The authors should describe the experimental methods in detail, including the HE staining method. The authors should also include the scientific names of the cats used.

5.     Results, line 162-174: The reviewer is not sure if you are describing Figure 1a. It is not clear where the three distinctive cell populations are indicated in Figure 1a. The reviewer does not even know where the stromal cells are in Figure 1a, and which are multinucleated giant cells. The author should mention multinucleated giant cells, macrophages, tumors, etc. in linkage with the figures.

6.     Line 178; The purpose of the experiment is not clear. The authors should state the purpose or goal of the experiments.

7.     Lines 178 and 180; The reviewers could not confirm whether nuclear or cytoplasmic localization was demonstrated. The authors need to clarify the localization of each marker protein by magnification and arrow pointing.

8.     Figure 1: It is not clear where the antibody-positive granules are located. The authors should indicate the precise position of the antibody-positive granules in the magnified Figures with arrows. The authors should also clearly indicate in the Figure 1 where the tissue is in the histological section images.

9.     The authors should include scale bars in all of Figure 1.

10.  The authors should mention the criteria for “+”, “-”, and “+/-” in Table 3 in the Mate & Meth.

11.  It is not known if the antibodies used are specific to cats. If the antibodies are of human origin, do you think they react correctly with the antigens in the cat's tissue? Authors should analyze the expression of markers by various methods.

12.  Please review the English.

Comments on the Quality of English Language

Please review the English.

Author Response

Comments and Suggestions for Authors

In this manuscript titled “Immunohistochemical characterization of feline giant cell tumor of bone (GCTb): what we know and what we can learn from human counterpart”, the authors analyze the expression of human GCTb marker proteins. This study of GCTb in cats provided interesting data.

The data presented in this study are quite preliminary. The experimental results presented in this manuscript are not clearly described and it is unclear whether conceptual advances can be achieved. Overall, the manuscript is too preliminary to be considered for publication. 

Comments:

  1. Line 25-26: The authors should select and list GCTb marker proteins. Ki-67 is used as a proliferation marker for GCTb and should be removed.

Author(s) response: We appreciate the reviewer's suggestion. The selection of GCTb marker proteins has been included in the manuscript, and Ki-67 was specifically used as a proliferation marker to indicate proliferative activity, as is commonly done in the human counterpart. We believe this marker is valuable in helping to differentiate GCTb from osteosarcomas. Therefore, we respectfully request to retain Ki-67 in the study for its relevance to the context of our analysis.

  1. Line 26-30: The study merely observed the expression of marker proteins in specific tissues. The reviewer does not know why that is sufficient to conclude that they are comparable to humans. The authors should logically describe what they found in this study.

Author(s) response: We appreciate the reviewer's suggestion. The conclusion stated in the manuscript only refer to the cell populations characterizing these lesions which are overlapping; larger case series complete with clinical information are needed to validate the model.

  1. Intro: The reasons for studying GCTb in cats with a small number of cases are not compelling. The authors should focus their statement on this point.

Author(s) response: This has been added in the manuscript, underlying the importance of the description of these cases as the rarity of this condition makes case retrieval quite challenging.  A reference for other veterinary pathologists may be useful for the routine diagnostic activity.

  1. Mate & Meth, line 126-144: How did you obtain the cat samples? How was the tissue fixed and under what conditions? What type of equipment was used to prepare the sections? What kind of microscope did you use? The authors should describe the experimental methods in detail, including the HE staining method. The authors should also include the scientific names of the cats used.

Author(s) response: Details regarding the nature of the samples and some of the equipment used have been added to the manuscript. We have not elaborated on routine staining methods, such as H&E, as we believe this would not contribute additional critical information to the context of the study.

  1. Results, line 162-174: The reviewer is not sure if you are describing Figure 1a. It is not clear where the three distinctive cell populations are indicated in Figure 1a. The reviewer does not even know where the stromal cells are in Figure 1a, and which are multinucleated giant cells. The author should mention multinucleated giant cells, macrophages, tumors, etc. in linkage with the figures.

Author(s) response: We thank the reviewer for the comment. However, in pathology reports it would sound weird to indicate the different cellular populations, as this is a basic histopathology feature.

  1. Line 178; The purpose of the experiment is not clear. The authors should state the purpose or goal of the experiments.

 Author(s) response: The reason and the aim of this descriptive study is clearly stated in LINE 113-120.

  1. Lines 178 and 180; The reviewers could not confirm whether nuclear or cytoplasmic localization was demonstrated. The authors need to clarify the localization of each marker protein by magnification and arrow pointing.

Author(s) response: We thank the reviewer for the comment. We understand that probably the reviewer is not a pathologist. But, again, this is a basic skill in the interpretation of an immunohistochemical and we believe that it might sound redundant for readers, that are expected to be mainly pathologists. Moreover, adding non-useful arrows or indications on top of an image would hide details, resulting in a less clear image for the reader.

  1. Figure 1: It is not clear where the antibody-positive granules are located. The authors should indicate the precise position of the antibody-positive granules in the magnified Figures with arrows. The authors should also clearly indicate in the Figure 1 where the tissue is in the histological section images.

Author(s) response: We thank the reviewer for the comment. We understand that probably the reviewer is not a pathologist. But, again, this is a basic skill in the interpretation of an immunohistochemical and we believe that it might sound redundant for readers, that are expected to be mainly pathologists. Moreover, adding non-useful arrows or indications on top of an image would hide details, resulting in a less clear image for the reader.

  1. The authors should include scale bars in all of Figure 1.

Author(s) response: Details on the magnification of the pictures are present in the picture details. We have not elaborated on this as we believe this would not contribute additional critical information to the context of the study.

  1. The authors should mention the criteria for “+”, “-”, and “+/-” in Table 3 in the Mate & Meth.

Author(s) response: Clear indications were present in the M&M and are now added in table 3.

  1. It is not known if the antibodies used are specific to cats. If the antibodies are of human origin, do you think they react correctly with the antigens in the cat's tissue? Authors should analyze the expression of markers by various methods.

Author(s) response: For each antibody not previously tested in veterinary medicine, we obtained the amino acid (aa) sequence of the antigen against which the antibody was raised. We then performed a sequence alignment between the canine protein and the species in which the antibody was originally validated by the manufacturer. This analysis was conducted on all the antibodies used in this study using the free online tools available on www.uniprot.it.

Additionally, for each antibody we tested different control tissues and tumors and compared the labelling pattern with the images reported for healthy and pathological tissues provided on www.proteinatlas.com

  1. Please review the English.

Author(s) response: English has been reviewed.

Reviewer 4 Report

Comments and Suggestions for Authors

In this study, based on 3 cases, the authors perform the immunohistochemical characterization of feline giant cell tumor of bone (GCTb) and make a comparative approach with the human counterpart.

The manuscript is very well written, the different sections are balanced so that it is easy to read and understand. The authors discuss the results adequately and assume the main limit of the study. 

1. INTRODUCTION

Comment 1: Although very well written, for the type of work, it seems a little long to me, perhaps it could be shortened.

 2. MATERIALS AND METHODS

I suggest adding additional information. 

Comment 2: Please, mention that the staining used was H&E

Comment 3: What immunohistochemical kit/method/detection system and chromogen were used? (yes, later says DAB OD mean, but it should be explicitly stated in the M&M)

Comment 4: Please write how were the slides observed (light microscopy) and the photomicrographs obtained (digital camera).

 3. RESULTS

Comment 5: Like histological slides, were IHC slides also evaluated by three pathologists?

Comment 6: I suggest, Figure 1. Histological (H&E, x400) and immunohistochemical (x400) characteristics of feline GCTb (since the photomicrographs were all obtained with x400; or any other way the authors deem more appropriate but avoiding repeating IHC and x400).

Comment 7: Fig 1. Please insert some notations for mitotic figures and cell type, etc (arrow, arrowhead…)

Comment 8: Fig 1. Please insert counterstain with Hematoxylin (?)

Comment 9: Fig 1. I suggest insert “Positive immunoreactivity is indicated by brownish staining”

Author Response

Comments and Suggestions for Authors

In this study, based on 3 cases, the authors perform the immunohistochemical characterization of feline giant cell tumor of bone (GCTb) and make a comparative approach with the human counterpart.

The manuscript is very well written, the different sections are balanced so that it is easy to read and understand. The authors discuss the results adequately and assume the main limit of the study. 

  1. INTRODUCTION

Comment 1: Although very well written, for the type of work, it seems a little long to me, perhaps it could be shortened.

Author(s) response: We understand the reviewer’s point of view; however, our goal was also to gather information for the readers, given GCTB in is a rarely diagnosed tumor and not well-known amongst veterinary pathologists.

  1. MATERIALS AND METHODS

I suggest adding additional information. 

Comment 2: Please, mention that the staining used was H&E

Author(s) response: This has been added in the manuscript.

Comment 3: What immunohistochemical kit/method/detection system and chromogen were used? (yes, later says DAB OD mean, but it should be explicitly stated in the M&M)

Author(s) response: This has been added in the manuscript. . Positivity was detected through a solution of aminoethyl carbazole (AEC). The parameter “DAB OD mean” only refers to the standard name of the QuPath parameter to set for positive cell detection.

Comment 4: Please write how were the slides observed (light microscopy) and the photomicrographs obtained (digital camera).

Author(s) response: This has been added in the manuscript.

  1. RESULTS

Comment 5: Like histological slides, were IHC slides also evaluated by three pathologists?

Author(s) response: This has been added in the manuscript.

Comment 6: I suggest, Figure 1. Histological (H&E, x400) and immunohistochemical (x400) characteristics of feline GCTb (since the photomicrographs were all obtained with x400; or any other way the authors deem more appropriate but avoiding repeating IHC and x400).

Author(s) response: This has been modified in the manuscript.

Comment 7: Fig 1. Please insert some notations for mitotic figures and cell type, etc (arrow, arrowhead…)

Author(s) response: This has been added in the picture.

Comment 8: Fig 1. Please insert counterstain with Hematoxylin (?)

Author(s) response: This has been added in the picture.

Comment 9: Fig 1. I suggest insert “Positive immunoreactivity is indicated by brownish staining”

Author(s) response: This has been added in figure legend.

Round 2

Reviewer 2 Report

Comments and Suggestions for Authors

Dear authors

Thank you for your comments

At least, the authors should include (as suplementary material if you want) images corresponding to positive and negative contols for each antibody tested. Is the minimal requested for a validated IHQ procedure. 

Images for a negative and positive tissue controls, for each antibody

Author Response

The answers to comments are in the attached file, together with the required images.

Reviewer 3 Report

Comments and Suggestions for Authors
  1. Lines 178 and 180; The reviewers could not confirm whether nuclear or cytoplasmic localization was demonstrated. The authors need to clarify the localization of each marker protein by magnification and arrow pointing.

Author(s) response: We thank the reviewer for the comment. We understand that probably the reviewer is not a pathologist. But, again, this is a basic skill in the interpretation of an immunohistochemical and we believe that it might sound redundant for readers, that are expected to be mainly pathologists. Moreover, adding non-useful arrows or indications on top of an image would hide details, resulting in a less clear image for the reader.

  1. Figure 1: It is not clear where the antibody-positive granules are located. The authors should indicate the precise position of the antibody-positive granules in the magnified Figures with arrows. The authors should also clearly indicate in the Figure 1 where the tissue is in the histological section images.

Author(s) response: We thank the reviewer for the comment. We understand that probably the reviewer is not a pathologist. But, again, this is a basic skill in the interpretation of an immunohistochemical and we believe that it might sound redundant for readers, that are expected to be mainly pathologists. Moreover, adding non-useful arrows or indications on top of an image would hide details, resulting in a less clear image for the reader.

Reviewer Unless immunostaining is clearly indicated in the figure, the reviewer cannot be convinced of the author's claim. The presentation of Figure 1 should be reconsidered.

  1. The authors should include scale bars in all of Figure 1.

Author(s) response: Details on the magnification of the pictures are present in the picture details. We have not elaborated on this as we believe this would not contribute additional critical information to the context of the study.

Reviewer: The inclusion of scale bars in the figures is important for understanding the size of the cells that populate the tissues and their intracellular structures. The authors should reconsider the inclusion of scale bars in the figures.

Author Response

We thank the reviewer for his/her work. The point-by-point answer is attached as a Word file.
